# Association of SNPs within *TMPRSS6* and *BMP2* genes with iron deficiency status in Saudi Arabia

**Osama M. Al-Amer[1,2]\*, Atif Abdulwahab A. Oyouni[2,3], Mohammed Ali Alshehri[2,3], Abdulrahman Alasmari[2,3], Othman R. Alzahrani[2,3], Saad Ali S. Aljohani[4], Noura Alasmael[5], Abdulrahman Theyab[6], Mohammad Algahtani[6], Hadeel Al Sadoun[7], Khalaf F. Alsharif[8], Abdullah Hamad[1], Wed A. Abdali[9], Yousef MohammedRabaa Hawasawi[9,10]\***

1 Department of Medical Laboratory Technology, Faculty of Applied Medical Sciences, University of Tabuk, Tabuk, Kingdom of Saudi Arabia, 2 Genome and Biotechnology Unit, Faculty of Sciences, University of Tabuk, Tabuk, Kingdom of Saudi Arabia, 3 Department of Biology, Faculty of Sciences, University of Tabuk, Tabuk, Kingdom of Saudi Arabia, 4 Department of Basic Medical Sciences, Faculty of Medicine, Alrayan Colleges, Almadinah Almunawarah, Kingdom of Saudi Arabia, 5 King Abdullah University for Science and Technology, Thuwal, Kingdom of Saudi Arabia, 6 Department of Laboratory Medicine, Security Forces Hospital, Mecca, Kingdom of Saudi Arabia, 7 Department of Medical Laboratory Technology, Faculty of Applied Medical Sciences, King Abdulaziz University, Jeddah, Kingdom of Saudi Arabia, 8 Department of Clinical Laboratory Sciences, College of Applied Medical Sciences, Taif University, Taif, Kingdom of Saudi Arabia, 9 Research Center, King Faisal Specialist Hospital and Research Center, Jeddah, Kingdom of Saudi Arabia, 10 College of Medicine, Al-Faisal University, Riyadh, Saudi Arabia

\* oalamer@ut.edu.sa (OMA); hyousef@kfshrc.edu.sa (YMH)

**Data Availability Statement:** All relevant data are within the manuscript.

**Funding:** This project was supported by the Deanship of Scientific Research (DSR), University

## Abstract

### Background

Globally, iron-deficiency anemia (IDA) remains a major health obstacle. This health condition has been identified in 47% of pre-school students (aged 0 to 5 years), 42% of pregnant females, and 30% of non-pregnant females (aged 15 to 50 years) worldwide according to the WHO. Environmental and genetic factors play a crucial role in the development of IDA; genetic testing has revealed the association of a number of polymorphisms with iron status and serum ferritin.

### Aim

The current study aims to reveal the association of *TMPRSS6* rs141312 and *BMP2* rs235756 with the iron status of females in Saudi Arabia.

### Methods

A cohort of 108 female university students aged 18–25 years was randomly selected to participate: 50 healthy and 58 classified as iron deficient. A 3–5 mL sample of blood was collected from each one and analyzed based on hematological and biochemical iron status followed by genotyping by PCR.

of Tabuk, Tabuk, under grant No. (S-1441-0080). This work was also supported by Taif University Researchers Supporting Program (project number: TURSP-2020/153), Taif University, Saudi Arabia.

**Competing interests:** The authors have declared that no competing interests exist.

**Abbreviations:** BMP2, bone morphogenic proteins 2; CI, confidence intervals; IDA, iron-deficiency anemia; OR, Odds ratios; SNPs, single nucleotide polymorphisms; *TMPRSS6*, transmembrane serine protease 6; WHO, World Health Organization.

## Results

The genotype distribution of *TMPRSS6* rs141312 was 8% (TT), 88% (TC) and 4% (CC) in the healthy group compared with 3.45% (TT), 89.66% (TC) and 6.89% (CC) in the iron-deficient group (*P = 0.492*), an insignificant difference in the allelic distribution. The genotype distribution of *BMP2* rs235756 was 8% (TT), 90% (TC) and 2% (CC) in the healthy group compared with 3.45% (TT), 82.76% (TC) and 13.79% (CC) in iron-deficient group (*P = 0.050*) and was significantly associated with decreased ferritin status (P = 0.050). In addition, *TMPRSS6* rs141312 is significantly (*P<0.001*) associated with dominant genotypes (TC+CC) and increased risk of IDA while *BMP2* rs235756 is significantly (*P<0.026*) associated with recessive homozygote CC genotypes and increased risk of IDA.

## Conclusion

Our finding potentially helps in the early prediction of iron deficiency in females through the genetic testing.

## 1 Introduction

Recently, iron-deficiency anemia (IDA) has become a global health problem mainly affecting women, children and older adults [1]. The World Health Organization (WHO) estimated in 2013 that 273,000 deaths worldwide were due to IDA [2]. The incidence of IDA is generally higher in low- and middle-income countries [3]. According to the WHO, the greatest numbers of pre-school children, pregnant and non-pregnant women suffering from IDA live in the Eastern Mediterranean countries which include those in the Middle East [1]. A high prevalence of IDA was reported in individuals with poor diets, especially in female students who usually skip breakfast [4]. Also, several recent studies have suggested that IDA is frequently reported in females and infants in Saudi Arabia [5–10]. Considering the detrimental long-term effects and high prevalence of IDA, more attention has been given to its prevention [11]. Current research has suggested that the underlying causes for the disease worldwide include both environmental and genetic factors.

The major causes of IDA in Saudi Arabia include inadequate consumption of Vitamin C, infrequent consumption of red meat and fish, and genetic/or family history of IDA [8, 10]. Several polymorphisms have been previously reported in the Saudi population as causative single nucleotide polymorphisms (SNPs) in diseases such as breast cancer [12–16], colon cancer [17–19], diffuse parenchymal lung disease [20], and acute myeloid leukemia [21, 22]. In addition, genetic risk factors for IDA and causative genes have also been identified including the *TMPRSS6* gene [23].

*TMPRSS6* is expressed predominantly in the liver and negatively regulates the synthesis of the universal iron governing hormone hepcidin and thus plays a crucial role in iron homeostasis [24]. As *TMPRSS6* plays a fundamental role in the development of IDA, several genome-wide association researchs have recognized common SNPs in *TMPRSS6* that effect iron status [25–28].

In addition to regulation by *TMPRSS6*, hepcidin production is delicately controlled by interleukin-6, by bone morphogenic proteins (BMPs) and by other iron-regulated pathways [29]. Bone morphogenic protein 2 (*BMP2*) induces hepcidin expression through the BMP co-receptor hemojuvenlin [30]. While hepcidin excess induces anemia, hepcidin deficiency

induces iron overload. Variants of the *BMP2* gene have previously been associated with hemochromatosis but not with IDA [25, 31, 32].

In a study previously undertaken of female students from the northern region of Saudi Arabia, the rs855791 SNP in *TMPRSS6* was found to be significantly associated with poor iron status [23]. The purpose of the current study is to investigate *TMPRSS6* rs1421312 and *BMP2* rs235756 SNPs and their association with iron deficiency status among female students at the University of Tabuk, Saudi Arabia. As far as we know, this is first article reporting on the association of polymorphism rs235756 of the *BMP2* gene with iron deficiency status.

## 2 Methods

### 2.1 Study design

The study was approved by the Research Ethics Committees of Tabuk University; a total of 108 female students aged 18–25 participated and signed informed consent forms. The students were separated into two groups: 50 healthy students (control) and 58 iron deficient students. Females, were excluded from the study. Ethical approval for the research was obtained from the University of Tabuk's Committee of Research Ethics and KFSHRC-Jed (IRB# 2018–36).

### 2.2 Biochemical finding

Blood samples of 3–5 mL were collected. The Beckman Coulter LH750 Analyzer (Beckman Coulter Inc., Miami, FL, USA) was utilized to determine levels of haemoglobin, red blood cells (RBCs), white blood cells (WBCs) and platelets. A useful machine (Hitachi, UK) was utilised to measure serum iron and ferritin. Concentration of ferritin<15 ng/ml and haemoglobin<12 g/dl were defined as iron deficiency anaemia, whereas levels of ferritin<15 ng/ml and haemoglobin>12 g/dl were defined as iron deficiency without anaemia [33].

### 2.3 DNA preparation

Genomic DNA was isolated via the QIAamp DNA kit (Qiagen, Valencia, CA, USA) according to the manufacturer's instructions. The DNA concentrations were measured by Nano-Drop 8000 spectrophotometer (Thermo Scientific, USA), and the DNA purity was evaluated by the absorbance ratios of A260/A280 and A260/A230. After that, all DNA samples were stored at −80 ˚C until use. PCR was conducted using a Taq DNA polymerase kit (Invitrogen) according to the manufacturer's instructions.

### 2.4 Genotyping

Allele-specific PCR and ARMS-PCR were used to detect *TMPRSS6* rs1421312 and *BMP2* rs235756, respectively using following primers:

To detect the *TMPRSS6* rs1421312 polymorphism (Fig 1) the following primers were used:

Forward1 (T allele): `AGCCAGTGGCTTAGCCATTCCA`

Reverse: `GTGGTGGCAGCATCAGAGCAAAG`

Forward2 (C allele): `AGCCAGTGGCTTAGCCATTCCG`

Reverse: `GTGGTGGCAGCATCAGAGCAAAG`

To detect the *BMP2* rs235756 polymorphism (Fig 2) the following primers were used:

Forward1: `CATAGAGCAGGGCCCAGAAGCT`

Reverse1: `TCAGGGTACTCACGAAAGAGAGA`

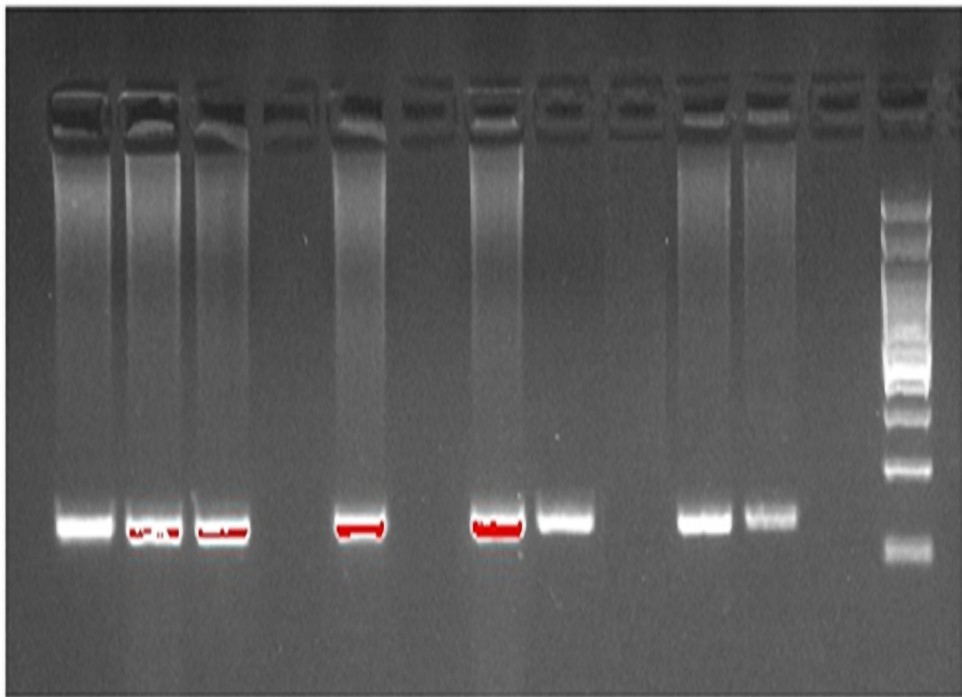

**Fig 1. Allele-specific PCR of *TMPRSS6* (rs1421312).**

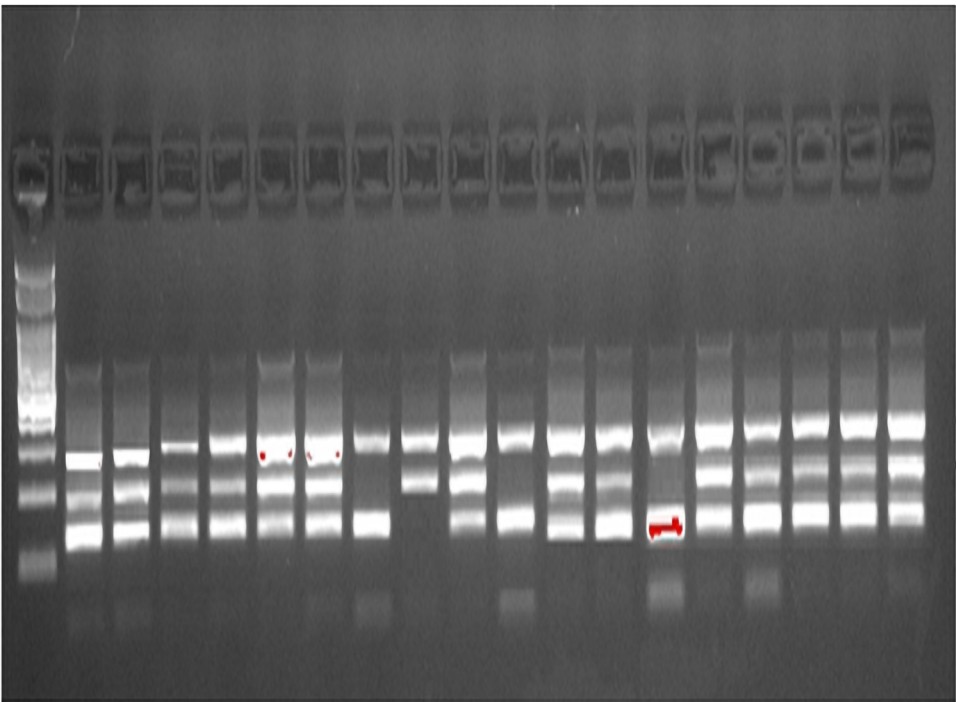

**Fig 2. ARMS-PCR of *BMP2* (rs235756).**

Forward2 (C allele): GAAGACTAAGAATTCTAGAATCCTCTCC

Reverse2 (T allele): AAGATTTTCCTTTGGGCACCTGTTGGT

Amplification was performed in a 25 μl reaction volume containing 50 ng genomic DNA, 0.4 μM of each primer, 250 μM dNTPs, 1.5 mM MgCl$_2$ and 1U Taq DNA polymerase. PCR amplification was performed with a 5-minute initial denaturation step at 95˚C followed by 30 seconds at 94˚C for denaturation, 30 seconds at 64˚C for annealing, 30 seconds at 72˚C for extension, and a final extension step at 72˚C for 5 minutes. The PCR was performed for 30 cycles. Products were separated using 2% agarose.

## 2.5 Statistical analysis

Statistical significance was determined using an χ$^2$ test or an independent student's t-test whenever appropriate. Results were considered statistically significant for the probability value (P) < 0.050. The odds ratios (OR) and 95% confidence intervals (CI) were calculated using the Chi-square test to determine the genetic variations in the two groups. Analyses were performed using SPSS version 16 (SPSS, Chicago, USA).

## 3 Results

### 3.1 Genotype distribution of *TMPRSS6* rs1421312 and *BMP2* rs235756

The genotype distribution of *TMPRSS6* rs1421312 was 8% (TT), 88% (TC) and 4% (CC) in the healthy group compared with 3.45% (TT), 89.66% (TC) and 6.89% (CC) in the iron-deficient group (Table 1). There was no significant statistical difference between the groups (*P = 0.492*) (Table 1).

In contrast, the genotype distribution of the *BMP2* rs235756 was 8% (TT), 90% (TC) and 2% (CC) in the healthy group compared with 3.45% (TT), 82.76% (TC) and 13.79% (CC) in the iron-deficient group (Table 2) which was statistically significant (*P = 0.050*).

### 3.2 Genotype distribution of *TMPRSS6* rs1421312 and *BMP2* rs235756 according to clinical parameters

To evaluate the distribution of *TMPRSS6* rs1421312 and *BMP2* rs235756 clinically, both were analyzed for iron, ferritin, haemoglobin, platelets, RBCs and WBCs (Tables 3 and 4, respectively) with the result that 78.7% of participants had high hemoglobin, 35% had high iron, 46% had high ferritin, 78% had high RBCs, 98% had high platelet counts and 74% had high WBCs while 22% of the students had low haemoglobin, 65% had low iron, 54% had low ferritin, 22% had low RBCs, 2% had low platelets and 26% had low WBCs.

**Table 1. Genotype distribution of *TMPRSS6* (rs1421312) gene polymorphism.**

|  | N | TT (%) | TC (%) | CC (%) | Chi-square | DF | p value |
|---|---|---|---|---|---|---|---|
| **Normal students** | 50 | 4 (8%) | 44 (88%) | 2 (4%) | 1.42 | 2 | 0.492 |
| **Iron deficient students** | 58 | 2 (3.45%) | 52 (89.66%) | 4 (6.89%) |  |  |  |

**Table 2. Genotype distribution of *BMP2* (rs235756) gene polymorphism.**

|  | N | TT | TC | CC | Chi-square | DF | p value |
|---|---|---|---|---|---|---|---|
| **Normal students** | 50 | 4 (8%) | 45 (90%) | 1 (2%) | 5.65 | 2 | 0.050 |
| **Iron deficient students** | 58 | 2 (3.45%) | 48 (82.76%) | 8 (13.79%) |  |  |  |

**Table 3. Genotype distribution of *TMPRSS6* (rs1421312) gene polymorphism with respect to clinical parameters.**

|  | N= | TT | TC | CC | X2 | Df | P value |
|---|---|---|---|---|---|---|---|
| **Hemoglobin** |  |  |  |  |  |  |  |
| **Normal** | 85 | 6 | 75 | 4 | 2.16 | 2 | 0.339 |
| **Abnormal** | 23 | 0 | 21 | 2 |  |  |  |
| **Iron** |  |  |  |  |  |  |  |
| **Normal** | 38 | 4 | 32 | 2 | 2.76 | 2 | 0.252 |
| **Abnormal** | 70 | 2 | 64 | 4 |  |  |  |
| **Ferritin** |  |  |  |  |  |  |  |
| **Normal** | 50 | 4 | 44 | 2 | 1.42 | 2 | 0.491 |
| **Abnormal** | 58 | 2 | 52 | 4 |  |  |  |
| **RBC** |  |  |  |  |  |  |  |
| **Normal** | 85 | 5 | 76 | 4 | 0.61 | 2 | 0.737 |
| **Abnormal** | 23 | 1 | 20 | 2 |  |  |  |
| **Platelets** |  |  |  |  |  |  |  |
| **Normal** | 106 | 6 | 94 | 6 | 0.25 | 2 | 0.883 |
| **Abnormal** | 02 | 0 | 2 | 0 |  |  |  |
| **WBC** |  |  |  |  |  |  |  |
| **Normal** | 80 | 6 | 69 | 5 | 2.61 | 2 | 0.271 |
| **Abnormal** | 28 | 0 | 27 | 1 |  |  |  |

Our data indicate that *TMPRSS6* rs1421312 was not significantly linked with decreased hemoglobin ($P = 0.339$), decreased serum iron ($P = 0.252$), decreased serum ferritin ($P = 0.491$), decreased RBCs ($P = 0.737$), decreased platelets ($P = 0.883$) or decreased WBCs ($P = 0.271$) (Table 3) while *BMP2* rs235756 was significantly associated with decreased serum ferritin ($P = 0.050$) but was not significantly associated with decreased hemoglobin ($P = 0.20$),

**Table 4. Genotype distribution of *BMP2* (rs235756) gene polymorphism with respect to clinical parameters.**

|  | N= | TT | TC | CC | X2 | Df | P value |
|---|---|---|---|---|---|---|---|
| **Hemoglobin** |  |  |  |  |  |  |  |
| **Normal** | 85 | 5 | 75 | 5 | 3.16 | 2 | 0.20 |
| **Abnormal** | 23 | 1 | 18 | 4 |  |  |  |
| **Iron** |  |  |  |  |  |  |  |
| **Normal** | 38 | 4 | 33 | 1 | 4.90 | 2 | 0.08 |
| **Abnormal** | 70 | 2 | 60 | 8 |  |  |  |
| **Ferritin** |  |  |  |  |  |  |  |
| **Normal** | 50 | 4 | 45 | 1 | 5.65 | 2 | 0.05 |
| **Abnormal** | 58 | 2 | 48 | 8 |  |  |  |
| **RBC** |  |  |  |  |  |  |  |
| **Normal** | 85 | 4 | 72 | 9 | 3.05 | 2 | 0.22 |
| **Abnormal** | 23 | 2 | 21 | 0 |  |  |  |
| **Platelets** |  |  |  |  |  |  |  |
| **Normal** | 106 | 6 | 91 | 9 | 0.33 | 2 | 0.85 |
| **Abnormal** | 02 | 0 | 2 | 0 |  |  |  |
| **WBC** |  |  |  |  |  |  |  |
| **Normal** | 80 | 4 | 70 | 6 | 0.50 | 2 | 0.78 |
| **Abnormal** | 28 | 2 | 23 | 3 |  |  |  |

decreased serum iron (*P = 0.08*), decreased RBCs (*P = 0.22*), decreased platelets (*P = 0.85*) or decreased WBCs (*P = 0.78*) (Table 4).

### 3.3 Association of *TMPRSS6* rs1421312 and *BMP2* rs235756 with iron deficiency anemia risk

*TMPRSS6* rs1421312 was genotyped in the healthy controls at 8%, 88%, and 4% for the TT, TC, and CC genotypes, respectively and at 0%, 95% and 5% in the iron-deficient group for the TT, TC, and CC genotypes, respectively (Table 5). In the healthy group, the T allele distribution was 52% while the C allele distribution was 48%. In contrast, the T allele distribution was 47% and the C allele distribution was 53% in the iron-deficient group. The OR 95% CI was 3.74(0.19–73.05) for the heterozygous TC genotype and 5.40(0.15–1.88) for the homozygous CC genotype.

There was a significant association of the heterozygous TC+ homozygous CC genotype with an increased risk for IDA (OR 95% CI; 3.77[0.19–73.5]), risk ratio: 1.41(1.20–1.65) and *P = 0.001*.

*BMP2* rs235756 was also genotyped in the iron-deficient group and was 5.3%, 73.7% and 21% for the TT, TC, and CC genotypes, respectively, whereas in the healthy group it was 6%, 94% and 0% for the TT, TC, and CC genotypes, respectively (Table 6). The T and C allele distributions were 42% for T and 58% for C in the iron-deficient group and 53% for T and 47% for C in the healthy group. The OR 95% CI was 29.3 (1.494–575.401) for the homozygous CC genotype. *BMP2* rs235756 was significantly (*P<0.026*) associated with increased risk for IDA in the female students in the study.

## 4 Discussion

In the past, iron deficiency was thought to be due to dietary and environmental factors. While this is partially true, advancements in technology and in the understanding of the underlying disorders of iron metabolism have revealed that genetic factors contribute heavily to the development of iron deficiency [34, 35]. At the population level, compelling evidence of geographic

**Table 5. Genotypes and allele frequencies of TMPRSS6 (rs1421312) polymorphism in normal subjects and in iron deficiency anemia group.**

| Genotypes | Normal subjects | | Anemia group | | OR (95% CI) | Risk Ratio (RR) | p-value |
|---|---|---|---|---|---|---|---|
| | (N = 50) | % | (N = 19) | % | | | |
| *TT* | 4 | 8% | 0 | 0% | 1(ref.) | 1(ref.) | |
| *TC* | 44 | 88% | 18 | 95% | 3.74 (0.19–73.05) | 1.40 (1.20–1.65) | 0.001 |
| *CC* | 2 | 4% | 1 | 5% | 5.40 (0.15–1.88.8) | 1.50 (0.67–3.33) | 0.32 |
| **Dominant** | | | | | | | |
| *TT* | 4 | 8% | 0 | 0% | 1(ref.) | 1(ref.) | |
| *TC+CC* | 46 | 92% | 19 | 100% | 3.77 (0.19–73.5) | 1.41 (1.20–1.65) | 0.001 |
| **Recessive** | | | | | | | |
| *TT+TC* | 48 | 96% | 18 | 95% | 1(ref.) | 1(ref.) | |
| *CC* | 2 | 4% | 1 | 5% | 1.33 (0.11.11.61) | 1.09 (0.48–2.46) | 0.83 |
| **Allele** | | | | | | | |
| *T* | 52 | 52% | 18 | 47% | 1(ref.) | 1(ref.) | |
| *C* | 48 | 48% | 20 | 53% | 1.20 (0.56–2.54) | 1.05 (0.85–1.29) | 0.62 |

ref = Reference

**Table 6. Genotypes and allele frequencies of BMP2 (rs235756) polymorphism in normal subjects and in iron deficiency anemia group.**

| Genotypes | Normal subjects | | Anemia group | | OR (95% CI) | Risk Ratio (RR) | p-value |
|---|---|---|---|---|---|---|---|
| | (N = 50) | % | (N = 19) | % | | | |
| *TT* | 3 | 6% | 1 | 5.3% | 1 (ref.) | 1 (ref.) | |
| *TC* | 47 | 94% | 14 | 73.7% | 0.89 (0.086 to 9.282) | 0.97 (0.543 to 1.742) | 0.92 |
| *CC* | 0 | 0% | 4 | 21% | 21 (0.639 to 690) | 7.0 (0.474 to 103.276) | 0.08 |
| Dominant | | | | | | | |
| *TT* | 3 | 6% | 1 | 5.3% | 1 (ref.) | 1 (ref.) | |
| *TC+CC* | 47 | 94% | 18 | 94.7 | 21 (0.639 to 690.030) | 1.03 (0.577 to 1.862) | 0.08 |
| Recessive | | | | | | | |
| *TT+TC* | 50 | 100% | 15 | 79% | 1(ref.) | 1(ref.) | |
| *CC* | 0 | 0% | 4 | 21% | 29.3 (1.494 to 575.401) | 7.65 (0.549 to 106.47) | 0.026 |
| Allele | | | | | | | |
| *T* | 53 | 53% | 16 | 42% | 1 (ref.) | 1 (ref.) | |
| *C* | 47 | 47% | 22 | 58% | 1.55 (0.729 to 3.296) | 1.12 (0.916 to 1.387) | 0.254 |

ref = Reference

differences in iron status supports the hypothesis of genetic variations across ethnicities [36], in particular among Asian and African populations [1, 37].

SNPs are well known to cause mutations in DNA structures that lead to susceptibility to various diseases [38, 39] and change amino acid sequencing in certain protein [40]. Polymorphisms in the *TMPRSS6* gene have a profound impact on iron metabolism. TMPRSS6 SNPs have been associated with IDA but causality has is not established [41]. Among the SNPs in *TMPRSS6* involved in IDA are rs869320724, rs767094129, rs786205059, rs137853119, and rs137853120 [42]. Gichohi-Wainaina et al. reported several differences in minor allele frequency in *TMPRSS6* SNPs that includes rs228919, rs4820268, rs228921, rs855791, rs2111833, rs575620, rs228918, and rs1421312 [43]. Accordingly, we hypothesized that *TMPRSS6* rs1421312 is associated with IDA in the female university students we studied in Saudi Arabia.

McLaren and colleagues studied SNP in *TMPRSS6* gene in four different groups and found no significant association in rs1421312 and decreases in serum ferritin concentration [44]. In our study, the same result was observed with respect to iron concentration, serum ferritin and serum iron, but we did find that *TMPRSS6* rs1421312 is significantly associated with dominant genotypes (TC+CC) and an increased risk for IDA for female university students in the north of Saudi Arabia.

In humans, the *BMP2* gene is located on chromosome 20 and is considered an excellent candidate for fibrodysplasia (myositis) [45]. Most hemochromatosis genetic base conditions are linked to *BMP2* as a result of homozygosity for the C*282Y* missense mutations that cause modification in the *HFE* gene. Milet and colleagues found a significant association between the rs235756 SNP of *BMP2* and the pre-therapeutic serum ferritin level [32] (corrected for multiple testing). Hepcidin excesses inducing anemia and hepcidin deficiencies inducing iron overloads have been associated with *BMP2*. Variants of *BMP2* have previously been associated with hemochromatosis, but not IDA [25, 31, 32]. Accordingly, we hypothesized that the SNP rs235756 on *BMP2* is potentially associated with IDA in female university students in Saudi Arabia.

In our study, we observed no significant association between increased risk of IDA and BMP2 rs236756 although there was a significant association between increased risk for IDA and serum ferritin. This result is consistent with the finding of Milet and co-authors [32]. In contrast, in 2012 An and colleagues found a direct association between SNP rs855791 in

*TMPRSS6* gene and increased risk of IDA but no association between SNP rs235756 in *BMP2* gene and increased risk [25]. This contradicts our finding of a direct association between increased risk of IDA and *BMP2* variant rs235756 in the recessive genotype (CC) as the RR was 7.65 (0.549–106.475) and. This could be due to other environmental factors and the different genetic make-ups of the populations studied. Although the study size was limited (108 participants), our result is still valid; confirmation with a greater number of participants is required.

## 5 Conclusions

The current study demonstrates the substantial roles for *TMPRSS6* polymorphic variant rs1421312 and *BMP2* polymorphic variant rs235756 in increased susceptibility for IDA in Saudi female students aged 18–25. We found that both *TMPRSS6* rs1421312 dominant genotype (TC+CC) and *BMP2* rs235756 recessive genotype (CC) are associated with an increased risk for IDA. *BMP2* rs235756 was found to be associated with ferritin though neither SNP showed a significant association with decreased serum hemoglobin, RBCs, platelets or WBCs. In future clinical settings, our finding potentially helps in the early prediction for iron deficiency in females through the genetic testing.

## Author Contributions

**Conceptualization:** Noura Alasmael.

**Data curation:** Osama M. Al-Amer, Othman R. Alzahrani, Abdullah Hamad, Yousef MohammedRabaa Hawasawi.

**Formal analysis:** Osama M. Al-Amer, Atif Abdulwahab A. Oyouni, Noura Alasmael.

**Funding acquisition:** Khalaf F. Alsharif.

**Investigation:** Osama M. Al-Amer, Mohammad Algahtani, Yousef MohammedRabaa Hawasawi.

**Methodology:** Osama M. Al-Amer, Atif Abdulwahab A. Oyouni, Saad Ali S. Aljohani, Hadeel Al Sadoun, Abdullah Hamad, Yousef MohammedRabaa Hawasawi.

**Project administration:** Othman R. Alzahrani, Yousef MohammedRabaa Hawasawi.

**Resources:** Atif Abdulwahab A. Oyouni, Abdulrahman Alasmari.

**Supervision:** Abdulrahman Alasmari.

**Validation:** Mohammed Ali Alshehri, Abdulrahman Theyab.

**Visualization:** Othman R. Alzahrani, Mohammad Algahtani.

**Writing – original draft:** Mohammed Ali Alshehri, Abdulrahman Theyab, Mohammad Algahtani, Khalaf F. Alsharif, Abdullah Hamad, Yousef MohammedRabaa Hawasawi.

**Writing – review & editing:** Abdulrahman Alasmari, Othman R. Alzahrani, Saad Ali S. Aljohani, Abdulrahman Theyab, Mohammad Algahtani, Khalaf F. Alsharif, Abdullah Hamad, Wed A. Abdali, Yousef MohammedRabaa Hawasawi.

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
