## [Decision Letter · Decision Letter 0]

7 May 2021

PONE-D-21-07480

Association of polymorphisms of genes BMP2 and TMPRSS6 with iron deficiency status in Saudi Arabia

PLOS ONE

Dear Dr. Hawsawi,

Thank you for submitting your manuscript to PLOS ONE. After careful consideration, we feel that it has merit but does not fully meet PLOS ONE’s publication criteria as it currently stands. Therefore, we invite you to submit a revised version of the manuscript that addresses the points raised during the review process.

Please pay particular attention to the criticisms raised by Reviewer2, including the problem with the English and the issue of genotype frequency discrepancy. 

We look forward to receiving your revised manuscript.

Kind regards,

Cinzia Ciccacci

Academic Editor

PLOS ONE

Journal Requirements:

4. Please include your tables as part of your main manuscript and remove the individual files. Please note that supplementary tables (should remain/ be uploaded) as separate "supporting information" files.

6. Thank you for stating the following in the Funding Section of your manuscript:

[Deanship of Scientific Research (DSR), University of Tabuk, Tabuk, Saudi Arabia.]

 [The author(s) received no specific funding for this work]

7. Thank you for submitting the above manuscript to PLOS ONE. During our internal evaluation of the manuscript, we found significant text overlap between your submission and the following previously published works, some of which you are an author.

 https://www.sciencedirect.com/science/article/abs/pii/S0378111920304364?via%3Dihub

Please revise the manuscript to rephrase the duplicated text, cite your sources, and provide details as to how the current manuscript advances on previous work. Please note that further consideration is dependent on the submission of a manuscript that addresses these concerns about the overlap in text with published work.

Reviewers' comments:

Reviewer's Responses to Questions

**Comments to the Author**

1. Is the manuscript technically sound, and do the data support the conclusions?

Reviewer #1: Partly

Reviewer #2: Partly

2. Has the statistical analysis been performed appropriately and rigorously? 

Reviewer #1: Yes

Reviewer #2: Yes

3. Have the authors made all data underlying the findings in their manuscript fully available?

Reviewer #1: No

Reviewer #2: Yes

4. Is the manuscript presented in an intelligible fashion and written in standard English?

Reviewer #1: No

Reviewer #2: No

5. Review Comments to the Author

Reviewer #1: Sample size is so small for PhD paper

On which criteria you classified the students as healthy and iron deficient

Have you made any questionnaire for participants? If yes provide it

Provide consent form of participants for provision of blood samples

Primers used for PCR are previously used or novelty designed

What is m in the first line of discussion?

Provide the form of ethical committee or institution review board of university

Reviewer #2: The current study tried to assess the effects of TMPRSS6 and BMP2 SNP on the risk of iron deficiency anaemia in adult females from Saudi Arabia. The authors have done a great job in planning and executing this study. Both the study design and the results are is clear laid out. However, the article would require a complete review of the English grammar to make it suitable for publication.

Here are my comments on the article:

1. The title should be reviewed for correctness. For example, “polymorphisms of genes TMPRSS6 and BMP2” may be written as SNPs within TMPRSS6 and BMP2 genes with iron deficiency….

2. Introduction is unnecessarily long. Can be shortened to begin by talking about IDA and exclude general description of iron.

3. Why was TMPRSS6 rs4820268 not included in the analysis. This SNP has been widely associated with low iron status in different global populations.

4. Further English editing might help to improve the grammar. For example, the last paragraph of the introduction “….to investigate polymorphism rs1421312 of the TMPRSS6 gene…” may be written as TMPRSS6 rs1421312.

5. In section 3.2, it is not clear what this 3-5cc means. It will be helpful to write this use SI units (mL for the blood volume).

6. In the results section, line 1, TMPRSS6 rs1421312 is wrongly written rs141312.

7. Genotype frequencies of the two SNPs in the study population differed widely from the global and all regional populations including Asian populations in the 1000 Genomes project. I recommend that the authors review their genotyping method to ensure that there were no errors.

8. In the section “Genotype distribution of TMPRSS6 rs1421312 and BMP2 rs235756 according to clinical parameters”, the description of the haematology and biochemical parameters using “normal” and “abnormal” should be high and low levels respectively.

9. In page 9, the sentence “Although the exact mechanism is unclear, compelling evidence emphasizes the role of TMPRSS6 polymorphisms in causing IDA”, needs to be reviewed. TMPRSS6 SNPs have been associated with IDA but causality has is not established. Similarly, the grammar in the first sentence of the last paragraph in page 9 needs to be reviewed.

10. In the last sentence on page 9, “TMPRSS6 rs141312” the SNP ID is incorrect.

11. The information about the location of BMP2 mentioned in the second sentence of page 10 needs to be reviewed. BMP2 is located in chromosome 20, see Here, but not Chr 2 as mentioned by the authors.

12. The last paragraph in page 10 needs to be reviewed. The sentence “we observed no significant association between increased risk for IDA and SNP rs235756 on BMP2” may be written as ….risk of IDA and BMP2 rs236756….

13. The conclusion that the results of this study “can be used as a predictive marker of IDA is Sauda Arabia” is premature. Further research on more genetic markers is needed to verify the clinical usability of this findings. These two SNPs are not sufficient to make this conclusion.

6. PLOS authors have the option to publish the peer review history of their article (what does this mean?). If published, this will include your full peer review and any attached files.

Reviewer #1: No

Reviewer #2: **Yes: **Dr Momodou W. Jallow

---

## [Author Response · Author response to Decision Letter 0]

17 Jul 2021

Letter of Response to Reviewer’s Comments

Dear chief editor for PLOS ONE Journal

Many thanks for your peer review and comments. With great pleasure, we would like to inform you that, the manuscript has been revised comprehensively and we address all the reviewers' comments. We hope our manuscript satisfied the editor and reviewer and be considered for publication in your decent journal. Our updated response to the reviewers as following: 

1- Response to Reviewer #1: 

Sample size is so small for PhD paper

Many thanks for the important comments and recommendations. In Saudi Arabia and due to the Arabian culture where females are totally separated from males in the university, it was very difficult to collect samples from the female students. We did our best to collect as much as we can and we finally obtained a total of 108 blood samples from female students. Moreover, for the iron deficient students, we excluded all students diagnosed with eating disorders, pregnant, and those taking nutritional medications or supplements which reduced the number of participants. 

On which criteria you classified the students as healthy and iron deficient

Many thanks for the important comments and recommendations. The criteria for the students as healthy and iron deficient were according to the WHO iron deficiency anemia assessment which was published in 2001. Levels of ferritin < 15 ng/ml and haemoglobin < 12 g/dl were defined as iron deficiency anaemia, whereas levels of ferritin < 15 ng/ml and haemoglobin > 12 g/dl were defined as iron deficiency without anaemia.

Have you made any questionnaire for participants? If yes provide it

Many thanks for this essential comment. However, because we measured the serum iron level there was no questionnaire for the participants.

Provide consent form of participants for provision of blood samples

Many thanks for this crucial comment. Consent form of participants for provision of blood samples is attached as requested. 

Primers used for PCR are previously used or novelty designed

Many thanks for this valuable point. The primers were novelty designed. 

What is m in the first line of discussion?

Many thanks for this precise notice. The m was a mistake from the proofreading and it has been removed. The sentence is " was thought to be due to dietary and environmental factors" and it has been highlighted in yellow. 

Provide the form of ethical committee or institution review board of university.

Many thanks for this essential point. Ethical approval (IRB #2018-36) is attached as requested. 

2- Response to Reviewer #2: 

 The current study tried to assess the effects of TMPRSS6 and BMP2 SNP on the risk of iron deficiency anaemia in adult females from Saudi Arabia. The authors have done a great job in planning and executing this study. Both the study design and the results are is clear laid out. However, the article would require a complete review of the English grammar to make it suitable for publication.

Many thanks for this and the manuscript has been sent for proofreading and English grammar was reviewed. 

1. The title should be reviewed for correctness. For example, “polymorphisms of genes TMPRSS6 and BMP2” may be written as SNPs within TMPRSS6 and BMP2 genes with iron deficiency….

Many thanks for this essential comment. The title has been changed according to your valuable advice, and highlighted in yellow.

2. Introduction is unnecessarily long. Can be shortened to begin by talking about IDA and exclude general description of iron.

This is a valid point and we agree with the reviewer. Therefore, the introduction has been shortened and focused on IDA. 

3. Why was TMPRSS6 rs4820268 not included in the analysis. This SNP has been widely associated with low iron status in different global populations.

Many thanks for this valid point. In our previous publication, we studied the TMPRSS6 rs4820268 SNPs in details. 

4. Further English editing might help to improve the grammar. For example, the last paragraph of the introduction “….to investigate polymorphism rs1421312 of the TMPRSS6 gene…” may be written as TMPRSS6 rs1421312.

Many thanks for this point and the manuscript has been sent for proofreading and English grammar were reviewed. The mentioned sentence has been corrected and highlighted in yellow.

5. In section 3.2, it is not clear what this 3-5cc means. It will be helpful to write this use SI units (mL for the blood volume).

Many thanks for this point and the 3-5cc was corrected to mL in the manuscript and highlighted in yellow in page 4. 

6. In the results section, line 1, TMPRSS6 rs1421312 is wrongly written rs141312.

Many thanks for this point. The SNP is now corrected as rs1421312 and highlighted in yellow. 

7. Genotype frequencies of the two SNPs in the study population differed widely from the global and all regional populations including Asian populations in the 1000 Genomes project. I recommend that the authors review their genotyping method to ensure that there were no errors.

Many thanks and the genotype frequencies of the two SNPs in the study population was reviewed and there was no error. 

8. In the section “Genotype distribution of TMPRSS6 rs1421312 and BMP2 rs235756 according to clinical parameters”, the description of the haematology and biochemical parameters using “normal” and “abnormal” should be high and low levels respectively.

Many thanks and the description of the haematology and biochemical parameters were changed and highlighted in yellow. 

9. In page 9, the sentence “Although the exact mechanism is unclear, compelling evidence emphasizes the role of TMPRSS6 polymorphisms in causing IDA”, needs to be reviewed. TMPRSS6 SNPs have been associated with IDA but causality has is not established. Similarly, the grammar in the first sentence of the last paragraph in page 9 needs to be reviewed.

Many thanks for this important point. The sentence was reviewed and corrected according to the reviewer advise and highlighted in yellow. 

10. In the last sentence on page 9, “TMPRSS6 rs141312” the SNP ID is incorrect.

Many thanks and the SNP has been corrected as rs1421312 and highlighted in yellow. 

11. The information about the location of BMP2 mentioned in the second sentence of page 10 needs to be reviewed. BMP2 is located in chromosome 20, see Here, but not Chr 2 as mentioned by the authors.

Many thanks for this important point, the zero was removed by mistake during the proofreading. Currently, it has been corrected to chromosome 20 and highlighted in yellow. 

12. The last paragraph in page 10 needs to be reviewed. The sentence “we observed no significant association between increased risk for IDA and SNP rs235756 on BMP2” may be written as ….risk of IDA and BMP2 rs236756….

Many thanks for this important point and the sentence has been corrected accordingly. 

13. The conclusion that the results of this study “can be used as a predictive marker of IDA is Sauda Arabia” is premature. Further research on more genetic markers is needed to verify the clinical usability of this findings. These two SNPs are not sufficient to make this conclusion.

This is a valid point and we totally agree with the reviewer and therefore the statement has been removed. 

Many thanks for your fairly and timely peer review along with valuable comments. We addressed all the highlighted reviewers' comments with a hope that, the revised manuscript will be considered for publication in your decent journal. 

Looking forward to hearing from you soon. 

Best regards

Yousef Hawsawi

---

## [Decision Letter · Decision Letter 1]

2 Sep 2021

PONE-D-21-07480R1

Association of SNPs within TMPRSS6 and BMP2 genes with iron deficiency status in Saudi Arabia

PLOS ONE

Dear Dr. Hawsawi,

Thank you for submitting your manuscript to PLOS ONE. After careful consideration, we feel that it has merit but does not fully meet PLOS ONE’s publication criteria as it currently stands. Therefore, we invite you to submit a revised version of the manuscript that addresses the points raised during the review process.

Please, revise the manuscript taking into account  the Reviewer 2 comments.

We look forward to receiving your revised manuscript.

Kind regards,

Cinzia Ciccacci

Academic Editor

PLOS ONE

Journal Requirements:

Additional Editor Comments (if provided):

Reviewers' comments:

Reviewer's Responses to Questions

**Comments to the Author**

1. If the authors have adequately addressed your comments raised in a previous round of review and you feel that this manuscript is now acceptable for publication, you may indicate that here to bypass the “Comments to the Author” section, enter your conflict of interest statement in the “Confidential to Editor” section, and submit your "Accept" recommendation.

Reviewer #1: All comments have been addressed

Reviewer #2: (No Response)

2. Is the manuscript technically sound, and do the data support the conclusions?

Reviewer #1: Partly

Reviewer #2: Yes

3. Has the statistical analysis been performed appropriately and rigorously? 

Reviewer #1: Yes

Reviewer #2: Yes

4. Have the authors made all data underlying the findings in their manuscript fully available?

Reviewer #1: Yes

Reviewer #2: Yes

5. Is the manuscript presented in an intelligible fashion and written in standard English?

Reviewer #1: Yes

Reviewer #2: Yes

6. Review Comments to the Author

Reviewer #1: Thanks for addressing all concerns. Refrain dual publication of article. try to meet the criteria of sample size

Reviewer #2: The authors have made significant improvements on the manuscript, and they have responded to the comments raised during the review.

However, there are minor issues that needs to be addressed to improve the quality of the article as follows:

1. Abstract

a. 3 – 5 cc in the methods section could be changes to 3 – 5mL.

2. Introduction section:

a. The sentence “….developing and low-income countries” could be change to “low- and middle-income countries”.

b. In line 3 of the third paragraph, the T in TMPRSS6 is not italised.

3. Results

a. In the last sentence of the last paragraph, it is not clear what the authors meant by … “associated with recessive genotypes”. The phrase may be removed.

4. Discussion section

a. The beginning of paragraph 3: “McLaren and his colleagues studied SNP in gene TMPRSS6 in four different ethnic groups…” should be revised. Also, the authors may consider removing the pronoun “his” when referring to other authors. Such sentences may be written as e.g. McLaren and colleagues studies.

b. In paragraph 4, line 4, the “H” in HFE should be italised.

c. Millet and his colleagues, see above comments

d. In line 6, it is not necessary to include the P value 0.002 in discussion here and in other places.

e. In paragraph 5, lines 4 – 5, the sentence “… direct association between TMPRSS6 and increase risk of IDA” should be revised. The association reported by these authors is between specific TMPRSS6 SNPs and IDA but not TMPRSS6 only.

Additional general comments

1. Table 5 need reformatting. In the Odd ration column, there should be a space between the OR and the confidence interval e.g 3.74(0.19-73.05) should be written as 3.74 (0.19-73.05).

2. There should be a footnote to explain what “ref.” means

3. Table 6 should be on one page. And see the above comments regarding footnotes and brakets.

7. PLOS authors have the option to publish the peer review history of their article (what does this mean?). If published, this will include your full peer review and any attached files.

Reviewer #1: No

Reviewer #2: **Yes: **Momodou W. Jallow, PhD.

---

## [Author Response · Author response to Decision Letter 1]

5 Sep 2021

Letter of Response to Reviewer’s Comments

Dear chief editor for PLOS ONE Journal

Many thanks for your distinguish peer review. With great pleasure, we would like to inform you that, the manuscript has been revised for the second round comprehensively and we address all the reviewers' comments. We hope our manuscript satisfied the editor and reviewers and be considered for publication in your decent journal. Our updated response to the reviewers as following: 

1- Response to Reviewer #1: 

Reviewer #1: Thanks for addressing all concerns. Refrain dual publication of article. try to meet the criteria of sample size

Many thanks for the positive comments and recommendations. We will try to meet the criteria of samples size in our next publication. We highly appreciate your valuable comments. 

2- Response to Reviewer #2: 

 Reviewer #2: The authors have made significant improvements on the manuscript, and they have responded to the comments raised during the review. 

Many thanks for the important comments and recommendations. It is highly appreciated.

However, there are minor issues that need to be addressed to improve the quality of the article as follows:

1. Abstract

a. 3 – 5 cc in the methods section could be changes to 3 – 5mL.

Many thanks for this important point. The “3-5 cc” in the methods section has been changed to 3-5 mL. 

2. Introduction section:

a. The sentence “….developing and low-income countries” could be change to “low- and middle-income countries”.

Many thanks for this important comments, the sentence has been changed to low- and middle-income countries. 

b. In line 3 of the third paragraph, the T in TMPRSS6 is not italised..

Many thanks for this precise comment and the T has been changed into italised.

3. Results

a. In the last sentence of the last paragraph, it is not clear what the authors meant by … “associated with recessive genotypes”. The phrase may be removed.

Many thanks for this essential comment. The phrase has been removed and the sentence changed to “associated with increased risk for IDA in the female students in the study”.

4. Discussion section

a. The beginning of paragraph 3: “McLaren and his colleagues studied SNP in gene TMPRSS6 in four different ethnic groups…” should be revised. Also, the authors may consider removing the pronoun “his” when referring to other authors. Such sentences may be written as e.g. McLaren and colleagues studies.

Many thanks for this important comment and the “McLaren and his colleagues studied SNP in gene TMPRSS6 in four different ethnic groups…” has been be revised and changed. The sentence McLaren and his colleagues has been corrected to McLaren and colleagues studies. 

b. In paragraph 4, line 4, the “H” in HFE should be italised.

Many thanks for this precise comment and the H has been changed into italised and highlighted in yellow

c. Millet and his colleagues, see above comments

Many thanks for this important comment. The Millet and his colleagues has been changed to Millet and colleagues and highlighted in yellow.

d. In line 6, it is not necessary to include the P value 0.002 in discussion here and in other places.

Many thanks for this important comment, the P values were removed. 

e. In paragraph 5, lines 4 – 5, the sentence “… direct association between TMPRSS6 and increase risk of IDA” should be revised. The association reported by these authors is between specific TMPRSS6 SNPs and IDA but not TMPRSS6 only..

Many thanks for this important comment, the sentence has been corrected and the SNPS were included. 

 

5. Additional general comments

1. Table 5 need reformatting. In the Odd ration column, there should be a space between the OR and the confidence interval e.g 3.74(0.19-73.05) should be written as 3.74 (0.19-73.05).

The table has been space was added and the column was updated.

2. There should be a footnote to explain what “ref.” means

The footnote was added.

3. Table 6 should be on one page. And see the above comments regarding footnotes and brakets.

The footnote was added and the space also was added. 

Many thanks for the fairly and timely peer review along with valuable comments. We addressed all the highlighted reviewers' comments with a hope that, the revised manuscript will be considered for publication in your decent journal. 

Looking forward to hearing from you soon. 

Best regards

Yousef Hawsawi

---

## [Editor Report · Decision Letter 2]

14 Sep 2021

Association of SNPs within TMPRSS6 and BMP2 genes with iron deficiency status in Saudi Arabia

PONE-D-21-07480R2

Dear Dr. Hawsawi,

We’re pleased to inform you that your manuscript has been judged scientifically suitable for publication and will be formally accepted for publication once it meets all outstanding technical requirements.

Kind regards,

Cinzia Ciccacci

Academic Editor

PLOS ONE
---

## [Editor Report · Acceptance letter]

5 Nov 2021

PONE-D-21-07480R2 

Association of SNPs within TMPRSS6 and BMP2 genes with iron deficiency status in Saudi Arabia

Dear Dr. Hawasawi:

I'm pleased to inform you that your manuscript has been deemed suitable for publication in PLOS ONE. Congratulations! Your manuscript is now with our production department. 

Kind regards, 

on behalf of

Dr. Cinzia Ciccacci 

Academic Editor

PLOS ONE